# Learning to Pass Expectation Propagation Messages

**Nicolas Heess**[*]
Gatsby Unit, UCL

**Daniel Tarlow**
Microsoft Research

**John Winn**
Microsoft Research

## Abstract

Expectation Propagation (EP) is a popular approximate posterior inference algorithm that often provides a fast and accurate alternative to sampling-based methods. However, while the EP framework in theory allows for complex non-Gaussian factors, there is still a significant practical barrier to using them within EP, because doing so requires the implementation of message update operators, which can be difficult and require hand-crafted approximations. In this work, we study the question of whether it is possible to automatically derive fast and accurate EP updates by learning a discriminative model (e.g., a neural network or random forest) to map EP message inputs to EP message outputs. We address the practical concerns that arise in the process, and we provide empirical analysis on several challenging and diverse factors, indicating that there is a space of factors where this approach appears promising.

## 1 Introduction

Model-based machine learning and probabilistic programming offer the promise of a world where a probabilistic model can be specified independently of the inference routine that will operate on the model. The vision is to automatically perform fast and accurate approximate inference to compute a range of quantities of interest (e.g., marginal probabilities of query variables). Approaches to the general inference challenge can roughly be divided into two categories. We refer to the first category as the "uninformed" case, which is exemplified by e.g. Church [4], where the modeler has great freedom in the model specification. The cost of this flexibility is that inference routines have a more superficial understanding of the model structure, being unaware of symmetries and other idiosyncrasies of its components, which makes the already challenging inference task even harder.

The second category is what we refer to as the "informed" case, which is exemplified by (e.g. BUGS[14], Stan[12], Infer.NET[8]). Here, models must be constructed out of a toolbox of building blocks, and a building block can only be used if a set of associated computational operations have been implemented by the toolbox designers. This gives inference routines a deeper understanding of the structure of the model and can lead to significantly faster inference, but the tradeoff is that efficient and accurate implementation of the building blocks can be a significant challenge. For example, EP message update operations, which are used by Infer.NET, often require the computation of integrals that do not have analytic expressions, so methods must be devised that are robust, accurate and efficient, which is generally quite nontrivial.

In this work, we aim to bridge the gap between the informed and the uninformed cases and achieve the best of both worlds by automatically implementing the computational operations required for the informed case from a specification such as would be given in the uninformed case. We train high-capacity discriminative models that learn to map EP message inputs to EP message outputs for each message operation needed for EP inference. Importantly, the training is done so that the learned modules implement the same EP communication protocol as hand-crafted modules, so after the training phase is complete, we get a factor that behaves like a fast hand-crafted approximation that exploits factor structure, but which was generated using only the specification that would be

---

[*]The majority of this work was done while NH was visiting Microsoft Research, Cambridge.

given in the uninformed case. Models may then be constructed from any combination of these learned modules and previously implemented modules.

## 2  Background and Notation

### 2.1  Factor graphs, directed graphical models, and probabilistic programming

As is common for message passing algorithms, we assume that models of interest are represented as factor graphs: the joint distribution over a set of random variables $\boldsymbol{x} = \{x_1, \dots, x_D\}$ is specified in terms of non-negative *factors* $\psi_1, \dots, \psi_J$, which capture the relation between variables, and it decomposes as $p(\boldsymbol{x}) = 1/Z \prod_{j=1}^{J} \psi_j(\boldsymbol{x}_{\psi_j})$. Here $\boldsymbol{x}_{\psi_j}$ is used to mean the set of variables that factor $\psi_j$ is defined over and whose index set we will denote by $Scope(\psi_j)$. We further use $\boldsymbol{x}_{\psi_{j-i}}$ to mean the set of variables $\boldsymbol{x}_{\psi_j}$ excluding $x_i$. The set $\boldsymbol{x}$ may have a mix of discrete and continuous random variables and factors can operate over variables of mixed types. We are interested in computing marginal probabilities $p_i(x_i) = \int p(\boldsymbol{x}) d\boldsymbol{x}_{-i}$, where $\boldsymbol{x}_{-i}$ is all variables except for $i$, and where integrals should be replaced by sums when the variable being integrated out is discrete. Note that this formulation allows for conditioning on variables by attaching factors with no inputs to variables which constrain the variable to be equal to a particular value, but we suppress this detail for simplicity of presentation.

Although our approach can be extended to factors of arbitrary form, for the purpose of this paper we will focus on *directed factors*, i.e. factors of the form $\psi_j(x_{out(j)} \,|\, \boldsymbol{x}_{in(j)})$ which directly specify the (conditional) distribution (or density) over $x_{out(j)}$ as a function of the vector of inputs $\boldsymbol{x}_{in(j)}$ (here $\boldsymbol{x}_{\psi_j}$ is the set of variables $\{x_{out(j)}\} \cup \boldsymbol{x}_{in(j)}$). In a (unconditioned) directed graphical model all factors will have this form, and we allow $\boldsymbol{x}_{in(j)}$ to be empty, for example, to allow for prior distributions over the variables.

Probabilistic programming is an umbrella term for the specification of probabilistic models via a programming-like syntax. In its most general form, an arbitrary program is specified, which can include calls to a random number generator (e.g. [4]). This can be related to the factor graph notation by introducing *forward-sampling functions* $f_1, \dots, f_J$. If we associate each directed factor $\psi_j(x_{out(j)} \,|\, \boldsymbol{x}_{in(j)})$ with a stochastic forward-sample function $f_j$ mapping $\boldsymbol{x}_{in(j)}$ to $x_{out(j)}$ and then define the probabilistic program as the sequential sampling of $x_{out(j)} = f_j(\boldsymbol{x}_{in(j)})$ following a topographical ordering of the variables, then there is a clear association between directed graphical models and forward-sampling procedures. Specifically, $f_j$ is a stochastic function that draws a sample from $\psi_j(x_{out(j)} \,|\, \boldsymbol{x}_{in(j)})$. The key difference is that the factor graph specification usually assumes that an analytic expression will be given to define $\psi_j(x_{out(j)} \,|\, \boldsymbol{x}_{in(j)})$, while the forward-sampling formulation allows for $f_j$ to execute an arbitrary piece of computer code. The extra flexibility afforded by the forward-sampling formulation has led to the popularity of methods like Approximate Bayesian Computation (ABC) [11], although the cost of this flexibility is that inference becomes less informed.

### 2.2  Expectation Propagation

Expectation Propagation (EP) is a message passing algorithm that is a generalization of sum-product belief propagation. It can be used for approximate marginal inference in models that have a mixed set of types. EP has been used successfully in a number of large-scale applications [5, 13], can be used with a wide range of factors and can support some programming language constructs like for loops and if statements [7]. For a detailed review of EP, we recommend [6].

For the purposes of this paper there are two important aspects of EP. First, we use the common variant where the posterior is approximated as a fully factorized distribution (except for some homogeneous variables which we treat as a single vector-valued variable) and each variable then has an associated type, $type(x)$, which determines the distribution family used in its approximation. The second aspect is the form of the message from a factor $\psi$ to a variable $i$. It is defined as follows:

$$m_{\psi i}(x_i) = \frac{\text{proj}\left[\int \psi(x_{out} \,|\, \boldsymbol{x}_{in}) \left(\prod_{i' \in Scope(\psi)} m_{i'\psi}(x_{i'})\right) d\boldsymbol{x}_{\psi - i}\right]}{m_{i\psi}(x_i).} \tag{1}$$

The update has an intuitive form. The proj operator ensures that the message being passed is a distribution of type $type(x_i)$ – it only has an effect if its argument is outside the approximating family used for the target message. If the projection operation (proj $[\cdot]$) is ignored, then the $m_{i\psi}(x_i)$

term in the denominator cancels with the corresponding term in the numerator, and standard belief propagation updates are recovered. The projection is implemented as finding the distribution $q$ in the approximating family that minimizes the $KL$-divergence between the argument and $q$: $\text{proj}\,[p] = \text{argmin}_q\, KL(p||q)$, where $q$ is constrained to be a distribution of $type(x_i)$. Multiplying the reverse message $m_{i\psi}(x_i)$ into the numerator before performing the projection effectively defines a "context", which can be seen as reweighting the approximation to the standard BP update, placing more importance in the region where other parts of the model have placed high probability mass.

## 3 Formulation

We now present the method that is the focus of this paper. The goal is to allow a user to specify a factor to be used in EP solely via specifying a forward sampling procedure; that is, we assume that the user provides an executable stochastic function $f(\boldsymbol{x}_{in})$, which, given $\boldsymbol{x}_{in}$ returns a sample of $x_{out}$. The user further specifies the families of distributions with which to represent the messages associated with the variables of the factor (e.g. Discrete, Gamma, Gaussian, Beta). Below we show how to learn fast EP message operators so that the new factor can be used alongside existing factors in a variety of models.

**Computing Targets with Importance Sampling**   Our goal is to compute EP messages from the factor $\psi$ that is associated with $f$, as if we had access to an analytic expression for $\psi(x_{out}\,|\,\boldsymbol{x}_{in})$. The only way a factor interacts with the rest of the model is via the incoming and outgoing messages, so we can focus on this mapping and the resulting operator can be used in any model. Given incoming messages $\{m_{i\psi}(x_i)\}_{i\in Scope(\psi)}$, the simplest approach to computing $m_{\psi i}(x_i)$ is to use importance sampling. A proposal distribution $q(\boldsymbol{x}_{in})$ is specified, and then the approach is based on the fact that

$$\int \psi(x_{out}\,|\,\boldsymbol{x}_{in}) \left( \prod_{i'\in Scope(\psi)} m_{i'\psi}(x_{i'}) \right) d\boldsymbol{x}_{\psi} = \mathbb{E}_r \left[ \frac{\prod_{i'\in Scope(\psi)} m_{i'\psi}(x_{i'})}{q(\boldsymbol{x}_{in})} \right], \qquad (2)$$

where $r(\boldsymbol{x}) = q(\boldsymbol{x}_{in})\psi(x_{out}\,|\,\boldsymbol{x}_{in})$ can be sampled from by first drawing values of $\boldsymbol{x}_{in}$ from $q$, then passing those values through the forward-sampling procedure $f$ to get a value for $x_{out}$. To use this procedure for computing messages $m_{\psi i}(x_i)$, we use importance sampling with proposal distribution $r$. Roughly, samples are drawn from $r$ and weighted by $\frac{\prod_{i'\in Scope(\psi)} m_{i'\psi}(x_{i'})}{q(\boldsymbol{x}_{in})}$, then all variables other than $x_i$ are summed out to yield a mixture of point mass distributions over $x_i$. The $\text{proj}\,[\cdot]$ operator is then applied to this distribution. Note that a simple choice for $q(\boldsymbol{x}_{in})$ is $\prod_{i'\in in} m_{i'\psi}(x_{i'})$, in which case the weighting term simplifies to just be $m_{out\psi}(x_{out})$. Despite its simplicity, however, we found this choice to often be suboptimal. We elaborate on this issue and give concrete suggestions for improving over the naive approach in the experiments section.

**Generation of Training Data**   For a given set of incoming messages $\{m_{i\psi}(x_i)\}_{i\in Scope(\psi)}$, we can produce a target outgoing message using the technique from the previous section. To train a model to automatically compute these messages, we need many example incoming and target outgoing message pairs. We can generate such a data set by drawing sets of incoming messages from some specified distribution, then computing the target outgoing message as above.

---

**Algorithm 1** Generate training data

---
1: **Input:** $\psi, i$, specifying we are learning to send message $m_{\psi i}(x_i)$.
2: **Input:** $\mathcal{D}^m$ training distribution over messages $\{m_{i'\psi}(x_{i'})\}_{i'\in Scope(\psi)}$
3: **Input:** $q(\boldsymbol{x}_{in})$ importance sampling distribution
4: **for** $n = 1 : N$ **do**
5:      Draw $m_0^n(x_0),\dots,m_D^n(x_D) \sim \mathcal{D}^m$
6:      **for** $k = 1 : K$ **do**
7:          Draw $\boldsymbol{x}_{in}^{nk} \sim q(\boldsymbol{x}_{in}^{nk})$ then compute $x_{out}^{nk} = f(\boldsymbol{x}_{in}^{nk})$
8:          Compute importance weight $w_{nk} = \frac{\prod_{i'\in Scope(\psi)} m_{i'\psi}^n(x_{i'}^{nk})}{q(\boldsymbol{x}_{in}^{nk})}$.
9:      **end for**
10:     Compute $\hat{\mu}^n(x_i) = \text{proj}\left[\frac{\sum_k w^{nk}\delta(x_i)}{\sum_k w^{nk}}\right]$
11:     Add pair $(\langle m_0^n(x_0),\dots,m_D^n(x_D)\rangle, \hat{\mu}^n(x_i))$ to training set.
12: **end for**
13: Return training set.

---

**Learning**   Given the training data, we learn a neural network model that takes as input the sufficient statistics defining $\boldsymbol{m}^n = \left\{m_{i'\psi}^n(x_{i'})\right\}_{i'\in Scope(\psi)}$ and outputs sufficient statistics defining

the approximation $g(\boldsymbol{m}^n)$ to target $\hat{\mu}^n(x_i)$. For each output message that the factor needs to send, we train a separate network. The error measure that we optimize is the average $KL$ divergence $\frac{1}{N}\sum_{n=1}^{N} KL(\hat{\mu}^n||g(\boldsymbol{m}^n))$. We differentiate this objective analytically for the appropriate output distribution type and compute gradients via back-propagation.

**Choice of Decomposition Structure**   So far, we have shown how to incorporate factors into a model when the definition of the factor is via the forward-sample procedure $f$ rather than as an analytic expression $\psi$. When specifying a model, there is some flexibility in how this capability is used. The natural use case is when a model can mostly be expressed using factors that have analytic expressions and corresponding hand-constructed operator implementations, but when a few of the interactions that we would like to use are more easily specified in terms of a forward-sampling procedure or would be difficult to implement hand-crafted approximations for.

There is an alternative use-case, which is that even if we have analytic expressions and hand-crafted implementations for all the factors that we wish to use in a model, it might be that the approximations which arise due to the nature of message passing (that is, passing messages that factorize fully over variables) leads to a poor approximation in some block of the model. In this case, it may be desirable to collapse the problematic block of several factors into a single factor, then to use the approach we present here. If the new collapsed factor is sufficiently structured in a statistical sense, then this may lead to improved accuracy. In this view, the goal should be to find groups of modeling components that go together logically, which are reusable, and which define interactions that have input-output structure that is amenable to the learned approximation strategy.

## 4   Related Work

Perhaps the most superficially similar line of work to the approach we present here is that of inference machines and truncated belief propagation [2, 3, 10, 9], where inference is done via an algorithm that is structurally similar to belief propagation, but where some parameters of the updates are learned. The fundamental difference between those approaches and ours is how the learning is performed. In inference machine training, learning is done jointly over parameters for all updates that will be used in the model. This means that the process of learning couples together all factors in the model; if part of the model changes, the parameters of the updates must be re-learned. A key property of our approach is that a factor may be learned once, then used in a variety of different models without need for re-training.

The most closely related work is ABC-EP [1]. This approach employs a very similar importance sampling strategy but performs inference simply by sending the messages that we use as training data. The advantage is that no function approximator needs to be chosen, and if enough samples are drawn for each message update, the accuracy should be good. There is also no up-front cost of learning as in our case. The downside is that generation and weighting of a sufficient number of samples can be very expensive, and it is usually not practical to generate enough samples every time a message needs to be sent. Our formulation allows for a very large number of samples to be generated once as an up-front cost then, as long as the learning is done effectively, each message computation is much faster while still effectively drawing on a large number of samples. Our approach also opens up the possibility of using more accurate but slower methods to generate the training samples, which we believe will be important as we look ahead to applying the method to even more complex factors. Empirically we have found that using importance sampling but reducing the number of samples so as to make runtime computation times close to our method can lead to unreliable inference.

Finally, at a high level, our goal in this work is to start from an informed general inference scheme and to extend the range of model specifications that can be used within the framework. There is work that aims for a similar goal but comes from the opposite direction of starting with a general specification language and aiming to build more informed inference routines. For example, [15] attempts to infer basic interaction structure from general probabilistic program specifications. Also of note is [16], which applies mean field-like variational inference to general program specifications. We believe both these directions and the direction we explore here to be promising and worth exploring.

## 5   Experimental Analyses

We now turn our attention to experimental evaluation. The primary question of interest is whether given $f$ it is feasible to learn the mapping from EP message inputs to outputs in such a way that the learned factors can be used within nontrivial models. This obviously depends on the specifics of $f$ and the model in which the learned factor is used. We attempt to explore these issues thoroughly.

**Choice of Factors**   We made specific choices about which functions $f$ to apply our framework to. First, we wanted a simple factor to prove the concept and give an indication of the performance that we might expect. For this, we chose the sigmoid factor, which deterministically computes $x_{out} = f(x_{in}) = \frac{1}{1+\exp(-x_{in})}$. For this factor, sensible choices for the messages to $x_{out}$ and $x_{in}$ are Beta and Gaussian distributions respectively. Second, we wanted factors that stressed the framework in different ways. For the first of these, we chose a *compound Gamma* factor, which is sampled by first drawing a random Gamma variable $r_2$ with rate $r_1$ and shape $s_1$, then drawing another random Gamma variable $x_{out}$ with rate $r_2$ and shape $s_2$. This defines $x_{out} = f(r_1, s_1, s_2)$, which is a challenging factor because depending on the choice of inputs, this can produce a very heavy tailed distribution over $x_{out}$. Another challenging factor we experiment with is the product factor, which uses $x_{out} = f(x_{in,1}, x_{in,2}) = x_{in,1} \times x_{in,2}$. While this is a conceptually simple function, it is highly challenging to use within EP for several reasons, including symmetries due to signs, and the fact that message outputs can change very quickly as functions of message inputs (see Fig. 3).

One main reason for the above factor choices is that there are existing hand-crafted implementations in Infer.NET, which we can use to evaluate our learned message operators. It would have been straightforward to experiment with more example factors that could not be implemented with existing hand-crafted factors, but it would have been much harder to evaluate our proposed method. Finally, we developed a factor that models the throwing of a ball, which is representative of the type of factors that we believe our framework to be well-suited for, and which is not easily implemented with hand-crafted approximations.

For all factors, we use the extensible factor interface in Infer.NET to create factors that compute messages by running a forward pass of the learned neural network. We then studied these factors in a variety of models, using the default Infer.NET settings for all other implementation details, e.g. message schedules and other factor implementations. Additional details of the models used in the experiments can be found in the supplemental material.

**Sigmoid Factor**   For the sigmoid factor, we ran two main sets of experiments. First, we learned a factor using the methodology described in Section 3 and evaluated how well the network was able to reconstruct the training data. In Fig. 1 we show histograms of KL errors for the network trained to send forward messages (Fig. 1a) and the network trained to send backwards messages (Fig. 1b). To aid the interpretation of these results, we also show the best, median, and worst approximations for each. There are a small number of moderate-sized errors, but average performance is very good.

We then used the learned factor within a Bayesian logistic regression model where the output nonlinearity is implemented using either the default Infer.NET sigmoid factor or our learned sigmoid factor. The number of training points is given in the table. There were always 2000 data points for testing. Data points for training and testing were generated according to $p(y = 1|x) = sigmoid(w^T x)$. Entries of $x$ were drawn from $N(0, 1)$. Entries of $w$ were drawn from $N(0, 1)$ for all relevant dimensions, and the others were set to 0. Results are shown in Table 1, which appears in the Supplementary materials. Predictive performance is very similar across the board, and although there are moderately large KL divergences between the learned posteriors in some cases, when we compared the distance between the true generating weights and the learned posteriors means for the EP and NN case, we found them to be similar.

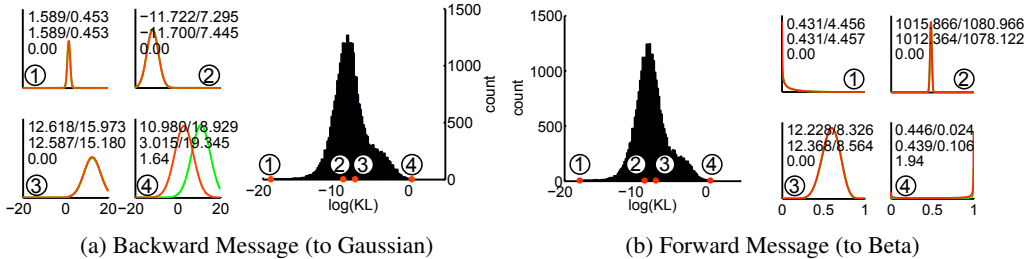

(a) Backward Message (to Gaussian)          (b) Forward Message (to Beta)

Figure 1: **Sigmoid factor**: Histogram of training KL divergences between target and predicted distributions for the two messages outgoing from the learned sigmoid factor (left: backward message; right: forward message). Also illustrated are best(1), median (2,3), and worst (4) examples. The red curve is the density of the target message, and the green is of the predicted message. In the inset are message parameters (left: Gaussian mean and precision; right: Beta $\alpha$ and $\beta$) for the true (top line) and predicted (middle line) message, along with the KL (bottom line).

**Compound Gamma Factor** The compound Gamma factor is useful as a heavy-tailed prior over precisions of Gaussian random variables. Accordingly, we evaluate performance in the context of models where the factor provides a prior precision for learning a Gaussian or mixture of Gaussians model. As before, we trained a network using the methodology from Section 3. For this factor, we fixed the value of the inputs $x_{in}$, which is a standard way that the compound Gamma construction is used as a prior. We experimented with values of $(3, 3, 1)$ and $(1, 1, 1)$ for the inputs. In both cases, these settings induce a heavy-tailed distribution over the precision.

We begin by evaluating the importance sampler. We first evaluate the naive choice for proposal distribution $q$ as described in Section 3. As can be seen in the bottom left plot of Fig. 2, there is a relatively large region of possible input-message space (white region) where almost no samples are drawn, and thus the importance sampling estimates will be unreliable. Here $shape_{in}$ and $rate_{in}$ denote the parameters of the message being sent from the precision variable to the compound Gamma factor. By instead using a mixture distribution over $q$, which has one component equivalent to the naive sampler and one broader component, we achieve the result in the top left of Fig. 2, which has better coverage of the space of possible messages. The plots in the second column show the importance sampling estimates of factor-to-variable messages (one plot per message parameter) as a function of the variable-to-factor message coming from the precision variable, which are unreliable in the regions that would be expected based on the previous plot. The third column shows the same function but for the learned neural network model. Surprisingly, we see that the neural network has smoothed out some of the noise of the importance sampler, and that it has extrapolated in a smooth, reasonable manner. Overlaid on these plots are the message values that were actually encountered when running the experiments in Fig. 8, which are described next.

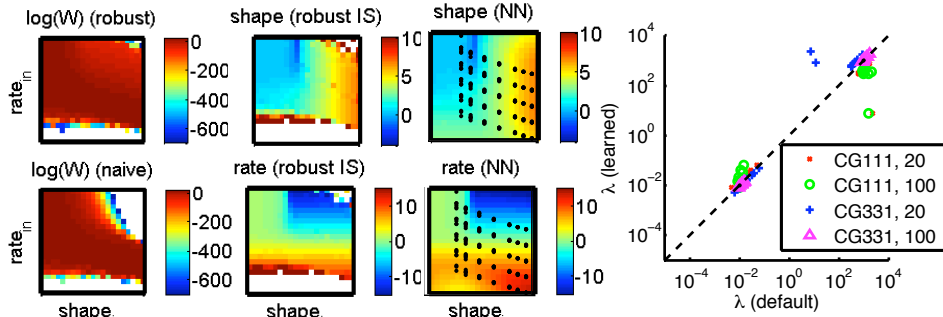

Figure 2: Compound Gamma plots. **First column:** Log sum of importance weights arising from improved importance sampler (top) and naive sampler (bottom) as a function of the incoming context message. **Second column:** Improved importance sampler estimate of outgoing message shape parameter (top) and rate parameter (bottom) as a function of the incoming context message. We show the sufficient statistics of the *numerator* of eq. 1. **Third column:** Learned neural network estimates for the same messages. Parameters of the variable-to-factor messages encountered when running the experiments in Fig. 8 are super-imposed as black dots. **Rightmost plot:** Precisions learned for mixture of Gaussians model with "learned" / standard Infer.NET ("default") factor for 20 and 100 datapoints respectively and true precisions: $\lambda_1 = 0.01$; $\lambda_2 = 1000$. Best viewed in color.

In the next experiments, we generate data from Gaussians with a wide range of variances, and we evaluate how well we are able to learn the precision as a function of the number of data points (x-axis). We compare to the same construction but using two hand-crafted Gamma factors to implement the compound Gamma prior. The plots in Fig. 8 in the supplementary material show the means of the learned precisions for two choices of compound Gamma parameters (top is $(3, 3, 1)$, bottom is $(1, 1, 1)$). Even though some messages were passed in regions with little representation under the importance sampling, the factor was still able to perform reliably.

We next evaluate performance of the compound Gamma factors when learning a mixture of Gaussians. We generated data from a mixture of two Gaussians with fixed means but widely different variances, using the compound Gamma prior on the precisions of both Gaussians in the mixture. Results are shown in the right-most plot of Fig. 2. We see that both factors sometimes under-estimate the true variance, but the learned factor is equally as reliable as the hand-crafted version. We also observed in these experiments that the learned factor was an order of magnitude faster than the built-in factor (total runtime was 11s for the learned factor vs. 110s for the standard Infer.NET construction).

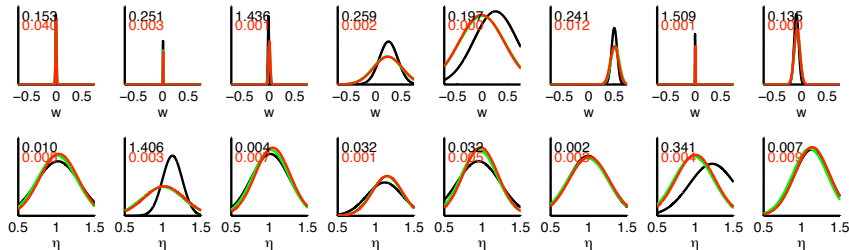

Figure 4: Learned posteriors from the multiplicative noise regression model. We compare the built-in factor's result (green) to our learned factor (red) and an importance sampler that is given the same runtime budget as the learned model (black). **Top row:** Representative posteriors over weights $w$. **Bottom row:** Representative posteriors over $\eta_n$ variables. Inset gives KL between built-in factor and learned factor (red) and IS factor (black).

**Product Factor**   The product factor is a surprisingly difficult factor to work with. To illustrate some of the difficulty, we provide plots of output message parameters along slices in input message space (Fig. 3). In our first experiment with the product factor, we build a Bayesian linear regression model with multiplicative output noise. Given a vector of inputs $x_n$, we take an inner product of $x_n$ with multivariate Gaussian variables $w$, then for each instance $n$ multiply the result by a random noise variable $\eta_n$ that is drawn from a Gaussian with mean 1 and standard deviation 0.1. Additive noise is then added to the output to produce a noisy observation $y_n$. The goal is to infer $w$ and $\eta$ values given $x$'s and $y$'s. We compare using the default Infer.NET product factor to using our learned product factor for the multiplication of $\eta$ and the output of the inner products. Results are shown in Fig. 4, where we also compare to importance sampling, which was given a runtime budget similar to that of the neural network.

In the second experiment with the product factor, we implemented an ideal point model, which is essentially a 1 latent-dimensional binary matrix-factorization model, using our learned product factor for the multiplications. This is the most challenging model we have considered yet, because (a) EP is known to be unreliable in matrix factorization models [13], and (b) there is an additional level of approximation due to the loopiness of the graph, which pushes the factor into more extreme

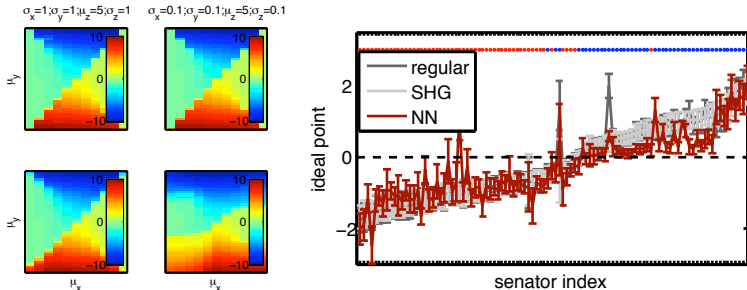

Figure 3: Message surfaces and failure case plot for the product factor (computing $z = xy$). **Left:** Mean of the factor to $z$ message as a function of the mean-parameters of the incoming messages from $x$ and $y$. Top row shows ground truth, the bottom row shows the learned NN approximation. **Right:** Posterior over the ideal-point variables for all senators (inferred std.-dev. is shown as error bars). Senators are ordered according to ideal-points means inferred with factor [13] (SHG). Red/blue dots indicate true party affiliation.

ranges, which it might not have been trained as reliably for and/or where importance sampling estimates used for generating training data are unreliable.

We ran the model on a subset of US senate vote records from the 112th congress.[1] We evaluated the model based on how well the learned factor version recovered the posteriors over senator latent factors that were found by the built-in product factor and the approximate product factor of [13]. The result of this experiment was that midway through inference, the learned factor version produced posteriors with means that were consistent with the built-in factors, although the variances were slightly larger, and the means were noisier. After this, we observed gradual degradation of the estimates for a subset of about 5-10% of the senators. By the end of inference, results had degraded significantly. Investigating the cause of this result, we found that a large number of zero-precision messages were being sent, which happens when the projected distribution has larger variance than

the context message. We believe that the cause of this is that as the messages in this model begin to converge, the messages being passed take on a distribution that is difficult to approximate (leading the neural network to underfit), that is different from the training distribution, or is in a regime where importance sampling estimates are noisy. In these cases, our KL-objective factors are overestimating the variance.

In some cases, these errors can propagate and lead to complete failure of inference, and we have observed this in our experiments. This leads to perhaps an obvious point, which is that our approach will fail when messages required by inference are significantly different from those that were in the training distribution. This can happen via the choice of too extreme priors, too many observations driving precisions to be extreme, and due to complicated effects arising from the dynamics of message passing on loopy graphs. We will discuss some possibly mitigating strategies in Section 6.

**Throwing a Ball Factor**    With this factor, we model the distance that a ball travels as a function of the angle, velocity, and initial height that it was thrown from. While this is also a relatively simple interaction conceptually, it would be highly challenging to implement it as a hand-crafted factor. In our framework, it suffices to provide a function $f$ that, given the angle, velocity, and initial height, computes and returns the distance that the ball travels. We do so by constructing and solving the appropriate quadratic equation. Note that this requires multiplication and trigonometric functions.

We learn the factor as before and evaluate it in the context of two models. In the first model, we have person-specific distributions over height (Gaussian), log slope (Gaussian) and the rate parameter (Gamma) of a Gamma distribution that deter-

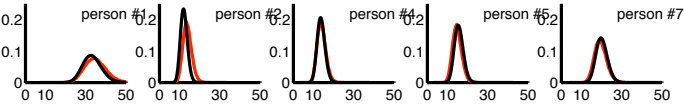

Figure 5: Throwing a ball factor experiments. True distributions over individual throwing velocities (black) and predictive distribution based on the learned posterior over velocity rates.

mines velocity. We then observe several samples (generated from the model) of noisy distances that the ball traveled for each person. We then use our learned factor to infer posteriors over the person-specific parameters. The inferred posteriors for several representative people are shown in Fig. 5. Second, we extended the above model to have the person-specific rate parameter be produced by a linear regression model (with exponential link function) with observed person-specific features and unknown weights. We again generated data from the model, observed several sample throws per person, and inferred the regression weights. We found that we were able to recover the generating weights with reasonable accuracy, although the posterior was a bit overconfident: true $(-.5, .5, 3)$ vs. posterior mean $(-.43, .55, 3.1)$ and standard deviations $(.04, .03, .02)$.

## 6  Discussion

We have shown that it is possible to learn to pass EP messages in several challenging cases. The techniques that we use build upon a number of tools well-known in the field, but the combination in this application is novel, and we believe it to have great practical potential.  Although we have established viability of the idea, in its current form it works better for some factors than others. Its success depends on (a) the ability of the function approximator to represent the required message updates (which may be highly discontinuous) and (b) the availability of reliable samples of these mappings (some factors may be very hard to invert). Here, we expect that great improvements can be made taking advantage of recent progress in uninformed sampling, and high capacity regression models. We tested factors with multiple models and/or datasets but this does not mean that they will work with all models, hyper-parameter settings, or datasets (we found varying degrees of robustness to such variations). A critical ingredient is here an appropriate choice of the distribution of training messages which, at the current stage, can require some manual tuning and experimentation. This leads to an interesting extension, which would be to maintain an estimate of the quality of the approximation over the domain of the factor, and to re-train the factor on the fly when a message is encountered that lies in a low-confidence region.  A second direction for future study, which is enabled by our work, is to add additional constraints during learning in order to guarantee that updates have certain desirable properties. For example, we may be able to ask the network to learn the best message updates subject to a constraint that guarantees convergence.

**Acknowledgements**: NH acknowledges funding from the European Community's Seventh Framework Programme (FP7/2007-2013) under grant agreement no. 270327, and from the Gatsby Charitable foundation.

## Footnotes

[1]Data obtained from `http://www.govtrack.us/`

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
