[Supplementary Material]

# Supplementary Materials for "Learning to Pass Expectation Propagation Messages"

## 1.1 Neural network architecture and optimization

We used standard neural networks to learn factor approximations. The exact network architecture varied from factor to factor and was chosen by experimentation. Networks had 3 or 4 hidden layers with 40 or 60 hidden units each. We used units with $\mathrm{tanh}$-activation function $g(x) = \tanh(x)$ for the hidden layers and nonlinearities as appropriate for each of the message parameters for the output units (e.g. a linear activation function $g(x) = x$ for a Gaussian mean parameter; and an exp-activation function $g(x) = \exp(x)$ for a precision parameter). Neural networks were optimized via batch scaled conjugate gradient descent.

## 1.2 Sigmoid factor

The factor graph of the model for Bayesian logistic regression is shown in Fig. 6. Table 1 compares results for logistic regression on several simulated datasets using the learned sigmoid factor with results obtained when using the standard Infer.NET implementation of the sigmoid factor. Table 2 shows similar results for the UCI Ionosphere dataset[2]

| Dataset | $D$ | $D_{rel}$ | $N$ | Prior var. | EP Test acc. | NN Test acc. | $KL(EP\|\|NN)$ |
|---|---|---|---|---|---|---|---|
| 1 | 10 | 4 | 500 | .1 | 0.7750 | 0.7745 | 1.9770 |
| 1 | 10 | 4 | 500 | 1 | 0.7750 | 0.7750 | 1.3247 |
| 2 | 50 | 10 | 500 | .1 | 0.7635 | 0.7635 | 0.4649 |
| 2 | 50 | 10 | 500 | 1 | 0.7660 | 0.7660 | 0.5641 |
| 4 | 50 | 40 | 500 | .1 | 0.8825 | 0.8820 | 0.6022 |
| 5 | 50 | 40 | 100 | 1 | 0.8065 | 0.8050 | 0.0847 |

Table 1: Results comparing Bayesian logistic regression with hand-crafted sigmoid factor (EP) versus learned sigmoid factor (NN). $D$: dimensionality of the data; $D_{rel}$: number of relevant dimensions; $N$:number of datapoints; *Prior var.*: Variance of prior on $w$ during inference; *EP Test acc.*: test accuracy of model using the default Infer.Net sigmoid-factor; *NN Test acc.*: test accuracy of model using the learned factor. $KL(EP\|\|NN)$: KL between posteriors inferred with the default factor and the learned factor.

| Prior var. | EP Test acc. | NN Test acc. |
|---|---|---|
| 0.025 | 0.8821 | 0.8793 |
| 0.050 | 0.8839 | 0.8800 |
| 0.100 | 0.8825 | 0.8751 |
| 0.250 | 0.87536 | 0.8746 |

Table 2: Results for Bayesian logistic regression on the UCI ionosphere dataset. Each dimension of the dataset was normalized to have zero mean and and a std-dev. of one. Numbers are cross validation results over 20 different 60/40 train/test splits of the dataset.

## 1.3 Compound Gamma factor

Figure 8 shows the results of inferring the precision of a Gaussian distribution using a compound Gamma prior (see main text for details).

Factor graphs of the mixture models leading to the results in Fig. 2 (right) are shown in Fig. 7. For these experiments we generated data from a mixture of Gaussians model with two zero-mean mixture components, one with low precision ($\lambda_1 = 0.01$), and one with high precision ($\lambda_2 = 1000$). We generated 20 or 100 datapoints and then inferred the posterior distribution over model parameters (prior probability $\pi$; precisions $\lambda_1, \lambda_2$) keeping the means fixed $\mu_1 = \mu_2 = 0$. The experiment was repeated 10 times.

Figure 6: Factor graph of the logistic regression model used in the experiments. Here, $y \in \{0,1\}$ and $y \sim \mathrm{Bernoulli}(p)$; $z = \mathbf{w}^T \mathbf{x}$; and $p = \sigma(z)$. $\mathbf{x}, \mathbf{w} \in \mathbb{R}^D$. The sigmoid factor is indicated in red. For $N$ see e.g. Table 1.

Figure 7: Factor graphs of the mixture of Gaussians models used in the compound Gamma experiments. Left: Mixture of two Gaussians model with learned compound Gamma factors (shown in red) that act as priors for the precisions of the mixture components. Right: Same model but with the "standard" Infer.NET implementation. The compound Gamma priors for the precisions are implemented via two Gamma distributions each: $\beta_B^i \sim \mathrm{Gamma}(\alpha_A^i, \beta_A^i)$ and $\lambda^i \sim \mathrm{Gamma}(\alpha_B^i, \beta_B^i)$. We use the gate representation for the mixture model.

## 1.4 Product factor

The models for multiplicative regression and the ideal point model are shown in Fig. 9.

Figure 8: Fitting a Gaussian with a compound Gamma prior on the precision. Each plot shows the true std-deviation (black) as well as one over the square root of the posterior mean over the precision as a function of the number of observed data points. For each number of observations we generated several datasets. Results for individual experiments are shown (markers) together with mean over experiments (lines). We compare the built-in factors (green curve) to the learned factor (red). (Top) $(3, 3, 1)$ compound Gamma prior on precisions. (Bottom) $(1, 1, 1)$ compound Gamma prior on precisions. In the right-most column examples of the full inferred posteriors over the precision are shown. The numbers in the inset are (Shape / rate (default), Shape / rate (learned), KL).

## 1.5 "Throwing a ball"-factor

The "ball-throwing" factor implements the following function:

$$
\begin{aligned}
c &\leftarrow 9.80665 \\
\theta &\leftarrow \tan^{-1}(\exp s) \\
v_y &\leftarrow v \sin \theta \\
v_x &\leftarrow v \cos \theta \\
v_c &\leftarrow v_y/(2c) \\
t_0 &\leftarrow v_c + \sqrt{v_c^2 + h/c} \\
d &\leftarrow v_x t_0.
\end{aligned}
$$

The factor graphs for the simple inference and for the regression experiment are shown in Figure 10.

Figure 5 in the main text shows the inferred predictive velocity for several subjects after 10 observed "attempts" per person and compares this inferred distribution over velocities to the subjects' true distributions. In Fig. 11 below we show directly the inferred posterior over the person-specific velocity rate parameter ($\beta_v$) for different numbers of observations together with an importance sampling (IS) estimate of this posterior (we perform IS in the full model). In order to increase the reliability of the IS estimate we assumed a higher level of observation uncertainty for this comparison than for the experiments in the main text. Figures 12 and 13 show corresponding results for the person specific log-slope mean ($\mu_s$) and height mean ($\mu_h$) parameters.

Figure 9: Factor graphs of models with product factor. (a) Regression model with multiplicative noise. Here, $z \sim \mathcal{N}(\mathbf{w}^T\mathbf{x}, 0.1)$ and $q = z\eta$ where $\eta \sim \mathcal{N}(1, 0.1)$. Gaussian noise $\epsilon \sim \mathcal{N}(0, \sigma_\epsilon^2)$ is added to obtain observation $y$ ($\sigma_\epsilon^2 = 1$). The product factor is shown in red. $N = 304$ in the experiments reported in the main text. (b) Ideal point model. $u$: ideal point; $b$: difficulty; $d$: discrimination; $v$: vote. In the experiments we had $\text{Bills} = 486$; $\text{Senators} = 102$ and we observed 5% of the votes.

Figure 10: Factor graphs of the models used in the ball-throwing simulations. (a) Simple inference experiment: There are $N$ subjects each of them throwing a ball $T$ times. The distance $d$ that the ball travels depends on the velocity $v$, (log-) slope $s$, and height $h$ at which the ball is released. These variables are drawn, for each attempt, from subject specific distributions. Log-slope and height are drawn from Gaussian distributions with subject specific means ($\mu_s$ and $\mu_h$) and precisions ($\lambda_s$ and $\lambda_h$). The velocity is drawn from a Gamma distribution with a subject specific rate $\beta_v$. (b) Velocity inference with regression. Same as (a) except that the subject specific velocity is determined via regression, with exponential link function, on a 3-dimensional subject specific feature vector $\mathbf{f}$, i.e. $\beta_v = \exp(\mathbf{w}^T \mathbf{f})$. For the experiments reported in the main text we have $T = 10$, $N = 9$.

Figure 11: Additional inference results for the ball-factor and comparison to importance sampling estimates. The figure shows the inferred posterior distribution over each subject's velocity rate parameter ($\beta_v$) given different number of observations (red trace). Also shown are estimates of the posteriors obtained by importance sampling in the full model (green traces). The true velocity rate parameter for each subject is shown by the black marker on the x-axis. As the number of observations increases the posterior inferred with the learned factor can become somewhat over-confident relative to the IS ground truth. This effect can be reduced by damping the the updates (dashed blue traces for 10, 20 observations). Damping is a well known technique for increasing the stability of EP algorithms.

Figure 12: Same format as Fig. 11 for the posterior over the person specific log-slope mean parameter ($\mu_s$). We note that inference for the mean log-slope can become unreliable for large numbers of observations, but that this tendency can be reduced by damping the updates (dashed blue traces).

Figure 13: Same format as Fig. 11 for the posterior over the person specific height mean parameter ($\mu_h$).

## Footnotes

[2]Bache, K. & Lichman, M. (2013). UCI Machine Learning Repository [http://archive.ics.uci.edu/ml]. Irvine, CA: University of California, School of Information and Computer Science.