[Reviews · NeurIPS 2013]

Submitted by Assigned_Reviewer_1

UPDATE: I acknowledge that I have read the author rebuttal.

The authors propose a method for learning a mapping from input messages to the output message in the context of expectation propagation. The method can be thought of as a sort of "compilation" step, where there is a one-time cost of closely approximating the true output messages using important sampling, after which a neural network is trained to reproduce the output messages in the context of future inference queries.

First, the authors should be commended for attacking a difficult and interesting problem. Learning to do inference has been a topic of increasing interest to our community, and it is good to see progress being made.

Clarity: The paper is well-written overall, with a few exceptions that I detail below.

Quality: The experiments are preliminary but convincing (at least as far as they go), although I would have liked a bit more detail on the setup (again, see below).

Originality: The approach is not particularly original; the contribution is more in bringing together several standard ideas. However, I think this is okay.

Significance: As stated before, the problem being addressed is an interesting one and I believe this lends to the significance. That being said, it would be far more exciting if it were possible to avoid the expensive "compilation" step; in many cases such a step is sufficiently expensive as to be infeasible, and I believe is where much of the difficulty lies. The authors address this issue partially by training each of the factors independently, but (i) there are sometimes factors with a large number of incoming messages, and (ii) I do not believe the authors have adequately demonstrated that training the factors independently does not lead to issues when we scale to large models with many approximately passed messages.

Other comments:
1. Phrases like "modeled as a Beta distribution" are confusing, you should make it clear that this refers to the approximating family being used in EP, as opposed to something else.
2. It's a bit unclear why the probabilistic programming formulation is necessary, this all seems
to only assume a directed graphical model, and this would simplify the exposition.
3. Directed graphical models are presented somewhat oddly, which confused me at first: why call it a "directed factor graph", and why the long-winded setup instead of just saying that it factors as $\prod_{i} p(x_i \mid x_{p(i)}$ or something similar? I think everyone knows that Bayes nets are.
4. It's hard to tell exactly what is going on in the experiments section, as you don't give any indication as to what exactly the model is, or how many nodes are in the Bayes net. Including model sizes and (if space permits) an explicit description of the model would improve clarity substantially.
Summary: Paper is well-written and addresses an interesting problem. Experimental results are good but preliminary; the paper could be improved by more convincingly demonstrating that the approximation scheme works in the context of a model with many factors using approximate messages (say >100 nodes with >20 approximate factors).

Submitted by Assigned_Reviewer_5

This paper proposes to learning expectation propagation (EP) message update operators from data that would enable fast and efficient approximate inference in situations where computing these operators is otherwise intractable.

- The authors address an interesting and relevant problem to the NIPS community and propose -- as far as I know -- a novel method of addressing intractable message update computations. The approach is reasonably straightforward: to use a more computationally demanding approximation process (importance-sampling driven Monte Carlo estimates) to generate data for use in learning a function approximation (such as a neural network) to provide fast and accurate approximate messages.

- One of the things I appreciated about this paper is its tone. The authors take pains to point out this approach is not a panacea and describe the circumstances where such a strategy would be effective. The experiments are designed to elucidate the strengths and weaknesses of the proposed approach and not necessarily restricted to trying to sell it.

- Regarding the exposition of the material: The paper was largely well written and easy to follow. However, I do have one major issue with the organization of the paper. Please refactor the paper and supplementary material to ensure that the paper stands on its own. Eg. remove descriptions of plots that appear in the supplementary material from the paper. This is essentially an abuse of the page limit on NIPS papers.

- Since computation time of the message updates is an important motivation of the proposed scheme, a careful and quantitative evaluation of it should be in the paper. It should include details of training times as well as "test" times.

- Another paper relevant to the idea of using function approximation to increase the speed of inference is:

Efficient learning of deep boltzmann machines. R Salakhutdinov, H Larochelle AISTATS 2010.

One thing that the above paper as well as refs [2], [3] and [11] have in common is their consideration of the approximation of inference in the context of parameter learning. Unless I missed something, the authors are rather silent on the issue of how to use their proposed method in the context of also learning parameters. Is this method practical in the inner-loop of parameter learning? Do the authors envision a kind of warm restart where relatively few training cycles would be required between learning steps? It seems like the use of computationally intensive methods such as importance sampling in the creation of the training data might render this method impractical as an inner-loop to parameter learning.

- The authors mention that one way their approach differs from these methods (mentioned above) is that they are exploiting modularity and that the learned message update operators could be re-used in other contexts. This is a good point that is likely mostly true, but isn't there some concern that the distribution over incoming messages could change from context to
context and that the learned operator might not be optimal for some of these contexts?

- Figure 2 needs to be much better explained with much more supporting text that it current has.
Summary: This paper proposes to learning expectation propagation (EP) message update operators from data that would enable fast and efficient approximate inference in situations where computing these operators is otherwise intractable. The authors present an interesting approach to an important problem.

Submitted by Assigned_Reviewer_6

This paper attacks the problem of computing the intractable low dimensional statistics in EP message passing by training a neural network. Training data is obtained using importance sampling and assuming that we know the forward model. The paper appears technically correct, honest about shortcomings, provides an original approach to a known challenge within EP and nicely illustrates the developed method in a number of well-chosen examples. The basic idea of the paper is refreshing, however, the paper does not entirely convince on the general applicability of the method. A convincing application is missing. This could for example by a novel EP application including the throwing a ball factor included as part of a bigger model or using the new factor calculation as part of an existing application showing no loss of robust or accuracy.
Summary: This paper attacks the problem of computing the intractable low dimensional statistics in EP message passing by learning the mapping from input statistics to output moments. The paper appears technically correct, honest about shortcomings, provides an original approach to a known challenge within EP and nicely illustrates the developed method in a number of well-chosen examples.
Author Feedback

Author rebuttal: We would like to thank all reviewers for their insightful and positive comments and suggestions.


Presentation (R1 & R5)

Thank you for the comments and suggestions related to presentation, and the points about organization are well-taken. We will simplify the presentation of background so that there is more space for plots in the main body and for additional details related to the experiments.


Additional experimental details and model complexity (R1, R5, R6)

In our experimental evaluation we chose several moderately complex models that are representative of models (and model components) used in practice, but that also highlight several upsides and downsides of the proposed method and allow for a careful analysis of its behavior.

We have not yet used different *types* of approximate factors in the same model. We will explore this in future work, although we emphasize that the models we consider represent a common case, in that there is often one particular type of factor (or possibly a small number of factor types) for which standard approximations are not available or that otherwise pose a performance bottleneck.

Nevertheless, almost all models in our experiments already contain multiple *instances* of the same approximate factor, e.g. several hundred in the case of logistic regression (100-500 depending on the dataset), and for regression with multiplicative noise (304 datapoints = 304 approximate factors). The model for Fig. 4 (ball throwing) comprises 90 instances of the learned factor.


Compute time (R5)

We do not claim that a significant improvement of test time performance can be obtained in all cases but it can be one advantage of our approach. For the compound gamma (CG) factor, where we observe and report such an improvement, the specific numbers are 11s (learned CG factor) and 110s (standard Infer.NET construction) for running the MoG experiments presented in the paper.

Our sigmoid factor performs very similar to the original Infer.NET factor: E.g. for the experiments reported in Table 1 in the Appendix we obtain 3.5s vs. 3.0s (learned vs. infer.net factor) for dataset 1 and 52s vs. 51s for dataset 4. The learned product factor is moderately faster: 25s (learned) vs. 35s (original Infer.NET) for the regression experiments with multiplicative noise. (In all cases these are total run times, for multiple repeats / cross-validation folds where appropriate.)

As learning the networks is a one-time up-front cost, the training procedure employed for the paper was chosen for ease of use rather than being optimized for runtime. The cost of data generation and network training ranges from hours to days depending on the factor.

We would like to emphasize that so far we have not explicitly optimized our approach with respect to either test or training compute time performance. We therefore expect that it will be possible to improve both substantially, e.g. using recent techniques and tools (e.g., GPUs) for improving neural network training, or via different choice of function approximator.


Parameter learning vs. inference (R5)

The view that we take on parameter learning (which we agree could be emphasized more strongly) is the Bayesian view in which parameters are just variables in the model, and the only task is inference. E.g., for the regression models, we are inferring posterior distributions over the weight variables in addition to all the other unobserved variables in the model. This "learned" posterior distribution is then used for prediction.

If we were to split parameter learning into an inner/outer loop structure, then our argument would be that we should learn the factors once, and then use the learned factors for all settings of the parameters encountered during learning. We did not explicitly do this, but we believe that inferring a posterior over weights (or for that matter, the parameters of a Gaussian mixture) is an equally strenuous of a test of the abilities of the learned factors to work in a range of parameter value regimes.



Modularity, factor re-use and context dependence (R5)

While we do expect the distribution of messages to change from context to context (e.g., as illustrated in the rightmost heat maps in Figure 1), the aim in this work was to choose a broad enough distribution of training messages and a powerful enough function approximator that the factor can operate successfully in the context of different models and hyperparameter settings, and we have shown in our experiments that this approach can work well in several cases.

Moving forward with this work, we agree with R5’s suggestion that more sophisticated methods of choosing message distributions could be helpful. For example, it would be interesting to explore an extension where the learned factors maintain knowledge about the distribution of messages that they have been trained for so far, and when they are asked to pass messages in regimes where they are not confident, they first generate more training data in that regime. We emphasize that this maintains the desirable modularity of our approach, i.e., the objective is fixed and defined locally per factor.